# PPARα, δ and FOXO1 Gene Silencing Overturns Palmitate-Induced Inhibition of Pyruvate Oxidation Differentially in C2C12 Myotubes

**DOI:** 10.3390/biology10111098

**Published:** 2021-10-25

**Authors:** Hung-Che Chien, Despina Constantin, Paul L. Greenhaff, Dumitru Constantin-Teodosiu

**Affiliations:** 1Queen’s Medical Centre, Division of Physiolgy, Pharmacology and Neuroscince, School of Life Sciences, University of Nottingham Medical School, Nottingham NG7 2UH, UK; chienhc@ndmctsgh.edu.tw (H.-C.C.); despina.constantin@nottingham.ac.uk (D.C.); Paul.Greenhaff@nottingham.ac.uk (P.L.G.); 2Department of Physiology and Biophysics, National Defense Medical Centre, Taipei 11490, Taiwan

**Keywords:** free fatty acids, metabolic inflexibility, muscle cell, fuel selection

## Abstract

**Simple Summary:**

Frequent high-dietary fat intake increases muscle lipid use and reduces muscle carbohydrate use, thereby inducing metabolic inflexibility. The latter term can be described as a poor muscle biochemical and molecular response to increased availability of insulin, which in the long term results in chronically excessive-high glucose levels in blood. Chronic hyperglycaemia is associated with many pathological conditions, including type 2 diabetes mellitus, which can cause severe health damages in humans. Here, we attempt to unravel the underlying mechanism and its associated factors behind the inhibition of muscle glucose use by a high-fat diet, thereby providing evidence for appropriate therapeutic intervention.

**Abstract:**

The molecular mechanisms by which free fatty acids (FFA) inhibit muscle glucose oxidation is still elusive. We recently showed that C2C12 myotubes treated with palmitate (PAL) presented with greater protein expression levels of PDK4 and transcription factors *PPARα* and *PPARδ* and lower *p*-*FOXO*/*t*-*FOXO* protein ratios when compared to control. This was complemented with the hallmarks of metabolic inflexibility (MI), i.e., reduced rates of glucose uptake, PDC activity and maximal pyruvate-derived ATP production rates (MAPR). However, the relative contribution of these transcription factors to the increase in PDK4 and reduced glucose oxidation could not be established. Therefore, by using a similar myotube model, a series of individual siRNA gene silencing experiments, validated at transcriptional and translation levels, were performed in conjunction with measurements of glucose uptake, PDC activity, MAPR and concentrations of metabolites reflecting PDC flux (lactate and acetylcarnitine). Gene silencing of *PPARα*, *δ* and *FOXO1* individually reduced PAL-mediated inhibition of PDC activity and increased glucose uptake, albeit by different mechanisms as only *PPARδ* and *FOXO1* silencing markedly reduced PDK4 protein content. Additionally, *PPARα* and *FOXO1* silencing, but not *PPARδ*, increased MAPR with PAL. *PPARδ* silencing also decreased FOXO1 protein. Since *FOXO1* silencing did not alter PPARδ protein, this suggests that *FOXO1* might be a *PPARδ* downstream target. In summary, this study suggests that the molecular mechanisms by which PAL reduces PDC-mediated glucose-derived pyruvate oxidation in muscle occur primarily through increased *PPARδ* and *FOXO1* mediated increases in PDK4 protein expression and secondarily through PPARα mediated allosteric inhibition of PDC flux. Furthermore, since *PPARδ* seems to control FOXO1 expression, this may reflect an important role for *PPARδ* in preventing glucose oxidation under conditions of increased lipid availability.

## 1. Introduction

Frequent nutrient overload, especially in the form of fat, contributes to the development of skeletal muscle metabolic inflexibility (MI), a state associated with mitochondrial dysfunction and a blunting of the switch in muscle fuel use from fat to glucose upon feeding or insulin-stimulated conditions [1,2]. The inability to switch fuel oxidation in response to changes in nutrient availability appears to be an important feature of chronic disorders such as obesity and type 2 diabetes [3]. Although metabolic inflexibility is most likely associated with reduced cellular glucose uptake [1], the downregulation of mitochondrial pyruvate dehydrogenase complex (PDC) activity by circulating fatty acids released from dietary fat is central to the induction of MI [4] due to PDC’s central role in glucose oxidative metabolism. Specifically, the rise in metabolic inflexibility appears to be related to an overexpression of the primary inhibitor of PDC, namely pyruvate dehydrogenase kinase isoform 4 (*PDK4*) in muscle and liver [4,5], organs that are quantitatively the most important sites of glucose disposal [6].

Through its exclusive role in committing glucose-derived pyruvate to acetyl-CoA in mitochondria, PDC plays a significant role in whole-body glucose homeostasis by enhancing muscle glucose oxidation response to insulin and exercise, which are the primary physiological activators of PDC [7,8,9]. However, the mechanisms controlling the reduction in glucose-derived pyruvate oxidation following chronic high-fat dietary intake are not fully understood, particularly at the level of PDC. Nevertheless, we have recently shown that C2C12 myotubes treated with palmitate (PAL) [10], which is the most prevalent saturated dietary free fatty acid [11], presented greater protein expression levels of PDK4, PPARα and PPARδ and a lower p-FOXO/t-FOXO ratio along with the hallmarks of metabolic inflexibility, i.e., reduced glucose uptake, PDC activity and maximal rates of pyruvate-derived ATP (MAPR) production compared to control.

Our previous work could not establish the relative contribution of *PPARα*, *PPARδ* and *FOXO1* transcription factors to the increase in PDK4 and blunted glucose oxidation. Therefore, by using a similar myotube model, a series of individual siRNA gene silencing experiments, validated at transcriptional and translation levels, was performed in conjunction with the quantification of cell glucose uptake, PDC activity and MAPR alongside measurements of metabolite concentrations reflecting PDC flux (lactate and acetylcarnitine). Additionally, we aimed to establish whether there are any interactions between PPARα, PPARδ and FOXO1 proteins.

## 2. Materials and Methods

### 2.1. Materials

Cell culture media (Dulbecco’s modified Eagle’s medium, phosphate-buffered saline and penicillin-streptomycin), sodium pyruvate, sodium palmitate and 2-deoxy-D-glucose were purchased from Sigma Aldrich (St. Louis, MO, USA). Fetal bovine serum (FBS), horse serum, cell extraction buffer and cocktail inhibitor were acquired from Life Technologies (Inchinnan, UK). Tritiated glucose, D-[3-3H], was obtained from Perkin Elmer (Boston, MA, USA). Mitochondrial ATP monitoring reagent was acquired from Biothema (Handen, Sweden). Protein assay reagents were purchased from BioRad (Copenhagen, Denmark). Anti-PDK4 (ab214938), anti-PPAR alpha (ab24509), anti-PPAR delta (ab23673), goat anti-Mouse (IRDye^®^ 800CW) (ab216772), goat Anti-Rabbit (IRDye^®^ 680RD) (ab216777) and anti-skeletal muscle alpha-actin (ab28052) antibodies were acquired from Abcam (Cambridge, UK). Anti-Phospho-FoxO1 (Ser256) (#9461) and Anti-FoxO1 (#2880) antibodies were purchased from Cell Signaling Technology (Danvers, MA, USA).

### 2.2. C2C12 Cell Culture, Differentiation into Myobutes and Small Interfering RNA (siRNA) Transfection

The C2C12 skeletal muscle cells used in the present experiments were acquired from ATCC (Manassas, VA, USA). The cells were cultured in collagen-coated 6-well plates and maintained in a growth media containing High Glucose Dulbecco’s Modified Eagle Media (DMEM), 10% Foetal Bovine Serum (FBS) and a 1% Pen/Strep. Once the myoblasts reached 80–100% confluency (after four days), they were differentiated into myotubes by using differentiating media containing high glucose DMEM, 2% Donor Horse Serum and 1% Pen/Strep. Two days into differentiation (day 6), the myotubes were transfected with 100 nM siRNA and 3 μL of Lipofectamine (ThermoFisher, Loughborough, UK) in an antibiotic-free medium in a 6-well plate according to the manufacturer’s instructions and allowed to grow for two additional days. At the end of day 8, the media were removed, and standard media (see above) with or without 500 μM palmitate were added to the cells, which were allowed to grow for a further 24 h. At the end of day 9, the media were removed, and the myotubes were harvested and used for different experimental measurements, as will be further described. The same concentrations of non-specific siRNA (scramble) with Lipofectamine were transfected in parallel plates as controls. The cells were constantly incubated at 37 °C in 5% CO_2_ throughout all phases and experiments.

### 2.3. Gene Expression Measurements

The cell gene expression levels were measured using real-time qPCR (ABI 7900HT sequence detection system, Applied Biosystems, Waltham, MA, USA). For *TaqMan probes*, we used *PDK4* (Mm01166879_m1); *HMBS* (Mm01143545_m1); *FOXO1* (Mm00490671_m1); *PPARα* (Mm00440939_m1); and *PPARδ* (Mm00440940_m1). For *siRNA probes,* we used negative silencer control (Nsi) (4390843), *PPARα*-s72005, *PPARδ*-s72009 and *FOXO1*-s80621.

### 2.4. Glucose Uptake Assay

Glucose uptake was determined as previously reported [12]. Briefly, C2C12 myotubes attached to the plate were incubated in 1 mL of media buffer supplemented with 138 mM NaCl, 1.85 mM CaCl_2_, 1.3 mM MgSO_4_, 4.8 mM KCl, 50 mM HEPES pH 7.4 and 0.2% (*w*/*v*) BSA for 2 h. Then, 250 μL of 2-deoxy-H^3^-glucose (1 mCi/mL) was added to the well, and the cells were incubated for 15 min. After incubation, the media were removed, and the well was washed three times with cold PBS buffer. After removing the last wash content, 500 μL of NaOH 50 mM and SDS 0.1% was added to solubilise the cells. A 500 μL aliquot of the solubilised cell mixture was removed and pipetted into a vial containing 5 mL of scintillation liquid, and radioactivity was measured with a scintillation counter (make and model).

### 2.5. Cell Metabolite Levels and PDC Activity Measurements

Medium lactate concentration was determined fluorometrically using a modified spectrophotometric method [13]. Acetylcarnitine (as an index of PDC flux; [14,15]) was measured in the cell extract using an enzymatic radioactive assay as previously described [16]. The cellular PDC activity was measured as described previously [7]. Briefly, C2C12 cells were extracted in a buffer containing sodium fluoride (NaF) and dichloroacetate (DCA). PDC activity in its dephosphorylated active form (PDCa) was assayed and expressed as the rate of acetyl-CoA formation (pmol acetyl-CoA/min/mg protein) at 37 °C.

### 2.6. Determination of Maximal Mitochondrial ATP Production Rates in Intact C2C12 Myotubes

Mitochondrial maximal ATP production rates (MAPR) from glucose-derived pyruvate were measured in C2C12 myotubes as described previously [17]. Of note, the MAPR measurements were conducted in the absence of palmitate, which was removed after cell pelleting and resuspension in the cell permeabilisation buffer, followed by resuspension in the ATP measuring buffer. Briefly, 3 μL of permeabilised cells was added to each well of a 96-well luminometer plate. Each well contained 200 µL of ATP monitoring reagent, 12.5 µL of 12 mM ADP and 35 µL of 375 mM pyruvate + 143 mM malate. Bioluminescence was recorded in duplicates continuously for 10 min (BMG LABTECH Ltd., Ortenberg, Germany), after which an injection of 150 pmoles ATP standard took place. The increase in the luminescence elicited by the ATP standard was used to calculate the values for absolute MAPRs. Subsequently, individual MAPRs were normalised for protein content by using the Bradford assay [18].

### 2.7. Western Blotting

The Near-Infrared (NIR) Western Blot Detection was employed [19], which is more dynamic, covers a more comprehensive linear range and has a higher sensitivity than compared to traditional chemiluminescent Western blotting. Briefly, C2C12 myotubes were lysed with a buffer containing a cocktail of protease and phosphatase inhibitors. The total protein concentration was determined using a protein assay (Bio-Rad, Hercules, CA, USA). In total, 25 μg of protein from each lysate was denatured, dissolved and separated with sodium dodecyl sulfate-polyacrylamide using gel electrophoresis (SDS-PAGE) in an XCell4 SureLock™ Midi-Cell system (Invitrogen, Inchinnan, UK). The separated proteins were transferred to a polyvinylidene fluoride (PVDF) or nitrocellulose membrane and blocked with 5% milk in Tris-buffered saline with Tween-20 (TBST) for 1 h at room temperature. After washing three times with TBST, the membrane was probed with the primary antibodies corresponding to the proteins of interest in TBST overnight at 4 °C. The following day, Licor IRDye 800CW (for α-actin) and 680RD (for PPARα, δ and FOXO1) fluorescent secondary antibodies were applied in TBST for 1 h at room temperature. The intensity of the protein bands was quantified using a near infra-red Licor imaging system (Odyssey^®^ CLx instrument; Cambridge, UK).

### 2.8. Statistical Analysis

All data are expressed as mean ± SEM of 6 individual experiments in each study group. The results for the cell glucose uptake are presented as the mean percentage (%) from the value of the first experiment in the glucose control group. This procedure provided a statistic (SEM) for the spreading of the dataset (including control) over day-to-day experimental runs. Two-way repeated-measures analysis of variance (ANOVA) was applied to identify treatment effects. When a significant F-ratio was obtained, a least significant difference (LSD) post hoc test was used to locate specific between treatment differences. Significance was set at the *p* < 0.05 level of confidence.

## 3. Results

### 3.1. Validation of PPARα, δ and FOXO1 siRNA Gene Silencing at Transcriptional and Translational Level in C2C12 Myotubes Treated with or without Palmitate (PAL)

The siRNA transfections were validated at transcriptional and translational levels by real-time PCR and Western blotting techniques, respectively.

Figure 1A shows that *PPARα* mRNA expression with or without PAL was significantly lower with *PPARα* siRNA treatment than their corresponding CON (both *p* < 0.01). Typical Western blots displayed in Figure 1B show that the mean PPARα protein level was significantly lower with *PPARα* gene silencing with or without PAL than CON (Figure 1C; both *p* < 0.01).

Figure 1D shows that *PPARδ* mRNA expression with or without PAL was significantly lower with *PPARδ* siRNA treatment than CON (both *p* < 0.01). Typical Western blots displayed in Figure 1E show that the mean PPARδ protein level was significantly lower with PPARδ gene silencing with or without PAL than CON (Figure 1F; both *p* < 0.01).

Figure 1G shows that *FOXO1* mRNA expression with or without PAL was significantly lower with *FOXO1* siRNA treatment than CON (both *p* < 0.01). Typical Western blots displayed in Figure 1H show that the mean total FOXO1 protein content with *FOXO1* siRNA intervention was significantly lower than CON with or without PAL (Figure 1I; both *p* < 0.01).

In summary, these data show that all siRNA transfections were successful.

### 3.2. Metabolic Outcomes after Individual PPARα, PPARδ and FOXO1 siRNA Gene Silencing in C2C12 Myotubes Treated with or without Palmitate (PAL)

#### 3.2.1. Glucose Uptake and Media Lactate Accumulation

Figure 2A shows glucose uptake expressed as % difference from CON after individual PPARα, PPARδ and FOXO1 gene silencing with or without PAL. In the absence of PAL, siRNA silencing did not affect glucose uptake in any group compared to CON.

However, following siRNA gene silencing, glucose uptake in all groups with PAL was significantly greater than CON with PAL (*p* < 0.01; all groups). The glucose uptake in CON with PAL was significantly lower than CON without PAL (*p* < 0.01).

Figure 2B shows media lactate concentration in C2C12 myotubes after individual *PPAR*α, *PPARδ* and *FOXO1* gene silencing with or without PAL. Irrespective of PAL’s absence or presence, media lactate concentration with PPARδ silencing was significantly lower than their corresponding CON (both *p* < 0.01). Although *FOXO1* silencing did not affect lactate levels without PAL, lactate levels were significantly lower than CON when PAL was present (*p* < 0.05).

#### 3.2.2. PDC Activity and Acetylcarnitine Concentration in C2C12 Myotubes Treated with or without Palmitate (PAL)

Figure 3A shows PDC activity in CON and after individual *PPAR*α, *PPARδ* and *FOXO1* gene silencing with or without PAL. In the absence of PAL, only siRNA *PPARδ* silencing significantly upregulated PDC activity (*p* < 0.01). With PAL, all gene silencing significantly upregulated PDC activity compared to CON (*p* < 0.01 in all conditions).

Figure 3B shows the naïve cellular acetylcarnitine concentration (used as an index of the flux through the PDC reaction) and cells with individual *PPARα*, *PPARδ* and *FOXO1* gene silencing with or without PAL. In the absence of PAL, acetylcarnitine concentration following PPARα gene silencing was significantly lower than CON (*p* < 0.01). In the presence of PAL, acetylcarnitine concentrations following PPARδ and FOXO1 gene silencing were significantly greater than CON + PAL (*p* < 0.05 in both conditions).

#### 3.2.3. Maximal Rates of Mitochondrial ATP Production (MAPR)-Based on Glucose-Derived Pyruvate Oxidation in C2C12 Myotubes Treated with or without Palmitate (PAL)

The results presented in Figure 4 show mean MAPR values in CON and after individual *PPARα*, *PPARδ* and *FOXO1* gene silencing with or without PAL. *PPARδ* silencing resulted in significantly lower MAPR than CON irrespective of the absence or the presence of PAL (*p* < 0.01 in both conditions), while *PPARα* and *FOXO1* gene silencing significantly increased MAPR relative to CON in the absence (*p* < 0.01 and *p* < 0.05, respectively) or the presence of PAL (*p* < 0.01 and *p* < 0.05, respectively).

### 3.3. Transcriptional and Translational Outcomes after Individual PPARα, PPARδ and FOXO1 siRNA Gene Silencing in C2C12 Myotubes Treated with or without Palmitate (PAL)

#### 3.3.1. PDK4 mRNA and Protein Expression

The effects of individual *PPARα*, *PPARδ* and *FOXO1* silencing on the mean *PDK4* gene expression in the absence (left panels) or the presence of palmitate (PAL; right panels) are presented in Figure 5A,D,G, respectively. The results show that *PPARα* silencing did not prevent upregulation of *PDK4* mRNA expression by PAL. However, *PPARδ* and *FOXO1* silencing prevented upregulation of *PDK4* mRNA by PAL (both *p* < 0.01).

Figure 5B,E,H show typical Western blots of PDK4 and their corresponding α-actin protein bands after *PPARα*, *δ* and *FOXO1* gene silencing interventions, respectively, with or without PAL. The mean PDK4 protein expression is presented in Figure 5C,F,I, respectively. Figure 5C shows that *PPARα* silencing did not prevent upregulation of PDK4 protein by PAL. However, both *PPARδ* and *FOXO1* silencing prevented upregulation of PDK4 protein by PAL (both *p* < 0.01).

#### 3.3.2. PPARδ and FOXO1 Translational and Post-Translational (Phosphorylation) Levels after PPARα siRNA Gene Silencing in C2C12 Myotubes Treated with or without Palmitate (PAL)

Figure 6A shows typical PPARδ and FOXO1 protein Western blots after *PPARα* silencing with or without PAL. *PPARα* siRNA gene silencing did not prevent the marked upregulation of PPARδ protein in all groups with PAL (Figure 6B). *PPARα* gene silencing with PAL significantly reduced both phosphorylated FOXO1 (Figure 6C, *p* < 0.05) and total-FOXO1 (Figure 6D, *p* < 0.01) protein levels, albeit by different magnitudes in such a manner that *p*-FOXO1/*t*-FOXO1 ratio increased significantly compared to PAL alone (Figure 6E, *p* < 0.01).

#### 3.3.3. PPARα and FOXO1 Translational and Post-Translational (Phosphorylation) Levels after PPARδ siRNA Gene Silencing in C2C12 Myotubes Treated with or without Palmitate (PAL)

Figure 7A shows typical Western blots after PPARδ gene silencing with or without PAL. *PPARδ* gene silencing with PAL decreased PPARα protein significantly compared to PAL alone (Figure 7B; *p* < 0.01). *PPARδ* gene silencing with PAL reduced both *p*-FOXO1 and *t*-FOXO1 protein expression significantly compared to PAL alone (Figure 7C,D, both *p* < 0.01), but the *p*-FOXO1/*t*-FOXO1 ratio remained unchanged from PAL alone (Figure 7E).

#### 3.3.4. PPARα and δ Translational Levels after FOXO1 siRNA Gene Silencing in C2C12 Myotubes Treated with or without Palmitate (PAL)

Figure 8A shows typical Western blots after FOXO1 silencing intervention with or without PAL. Figure 8B shows *FOXO1* gene silencing decreased PPARα protein expression with PAL compared to PAL alone (*p* < 0.01). However, the *FOXO1* gene silencing with PAL treatment did not change PPARδ protein expression (Figure 8C).

## 4. Discussion

The present study consolidated previous findings, which showed that gene and protein expressions of *PPARα δ* and *FOXO1* transcription factors in C2C12 skeletal muscle myotubes are markedly augmented when the cells are treated with PAL compared to the absence of PAL [10]. Additionally, these molecular outcomes were associated with hallmarks of MI/IR, i.e., lower rates of cellular glucose uptake, PDC activity and flux through PDC reaction and maximal glucose-derived pyruvate ATP production (MAPR).

In order to gain further mechanistic insight, we executed siRNA gene silencing of *PPARα*, *δ* and *FOXO1* transcription factors individually. All three transcription factors have been shown by us and others to be associated with an increase in PDK4 protein content in cells treated with palmitate [10,20,21,22]. Therefore, we hypothesized that their knockdown would overcome PAL induced inhibition of PDC activity and increase glucose-derived pyruvate oxidation.

It was found that *PPARα*, δ and *FOXO1* siRNA gene silencing reversed the PAL-mediated decline in PDC activity in all three instances. However, it appears that this was achieved through different mechanisms, as only *PPARδ* and *FOXO1* siRNA gene silencing reduced the expression of PDK4 protein markedly. Therefore, it seems that *PPARα* may downregulate glucose oxidation independently from the levels of PDK4 protein [23]. This latter finding bears resemblance to the result that overexpression of *PPARα* lowers glucose oxidation in rodent heart tissue by decreasing flux through PDC reaction exclusively via the allosteric inhibition of PDC by the products of FFA oxidation, rather than through levels of PDK4 protein [24]. Of note here, the *p*-FOXO1/*t*-FOXO1 ratio increased significantly after *PPARα* gene silencing with PAL in the present study (Figure 6E). Although *FOXO1* robustly upregulates muscle PDK4 protein during states associated with elevated circulating FFA levels (e.g., starvation) [21], conversely, an increase in *p*-FOXO1 (phosphorylated form), such as that observed after *PPARα* gene silencing, would induce inactivation of FOXO1 [25], thereby providing an even weaker molecular drive for downregulating PDK4 mRNA and protein (Figure 5A,C).

Our investigation also showed that *PPARα*, *PPARδ* and *FOXO1* silencing with PAL improved glucose uptake and glucose-derived pyruvate oxidation compared with naïve cells. Furthermore, *PPARδ* and *FOXO1* silencing with PAL lowered cell media lactate and increased acetylcarnitine concentrations, which collectively point towards improved flux through PDC reaction compared to *PPARα* silencing.

Usually, there is a positive association between the rate of flux through the PDC reaction and the magnitude of intracellular acetylcarnitine accumulation. When the rate of PDC mediated acetyl-CoA formation exceeds the rate of acetyl-CoA entry into the citric acid cycle, the excess acetyl-CoA is buffered by carnitine, which causes the cellular content of acetylcarnitine rise [14,15]. Collectively, these findings point to flux through PDC reaction after *PPARδ* and *FOXO1* silencing with PAL being significantly greater than after PPARα silencing. Therefore, the latter finding adds more weight to the contention that the effect of functional PPARδ and FOXO1 on PDC activity is more impactful than that of PPARα.

Additionally, *PPARα* and *FOXO1*, but not *PPARδ* silencing, rescued MAPR from the effects of PAL. It was also found that *PPARδ* silencing reduced FOXO1 protein, although FOXO1 silencing did not change PPARδ protein, suggesting that FOXO1 may potentially be a PPARδ downstream target.

Several previous studies showed that overexpression of *PPARδ* and *FOXO1* via agonism or intralipid infusion in in vivo models upregulate PDK4 gene or protein expression, thereby decreasing skeletal muscle PDC activity [26,27,28]. Conversely, it was shown here that *PPARδ* and *FOXO1* silencing downregulate PDK4 gene and protein expression and improved PDC activity and glucose-derived pyruvate oxidation with PAL. Among the genes investigated here, it was found that *PPARδ* silencing had the highest potency of all three for improving PDC activity (45%), suggesting that along with the well-established role of PPARδ in mobilising fat oxidation, it also has, although not uniquely, the ability to reduce glucose-derived pyruvate oxidation. After each gene silencing, the PDK4 protein expression levels recorded revealed that *PPARδ* and *FOXO1* genes, rather than *PPAR*α, mediated the rise in PDK4 protein with PAL, thereby confirming that PPARδ and FOXO1 transcription factors are de facto central modulators controlling PDC activity.

Since both *PPARδ* and *FOXO1* silencing resulted in lower PDK4 protein expression levels, this may signify similar roles for these two transcription factors in PDK4 modulation. However, since *PPARδ* silencing had the most significant recovery effect on PDC activity with PAL, this may suggest that PPARδ should also have had the most significant impact in modulation glucose-derived pyruvate oxidation out of any of the transcription factors investigated. However, our additional data contradict this. Thus, from a mitochondrial perspective, the MAPR responses with PPARδ silencing and PAL seemed to follow a different pattern from that of PPARα and FOXO1 silencing, which was to depress MAPR further rather than rescuing it. This finding may seem counterintuitive in hindsight also goes against a previous in vivo study, which showed that *PPARδ* agonism inhibited MAPR [29]. However, it is worth noting that PPARδ also controls the expression of skeletal muscle PGC-1*α* (peroxisome-proliferator-activated receptor γ activator-1α) [30,31]. PGC-1α is a co-transcriptional regulation factor that induces mitochondrial biogenesis by activating different transcription factors, including nuclear respiratory factor 1 and nuclear respiratory factor 2, which activate mitochondrial transcription factor A. The latter drives the transcription and replication of mitochondrial DNA (mtDNA) [32].

Given the above background information, it can be cautiously assumed that a decline of the mtDNA encoded mitochondrial proteins (*n* = 13, e.g., the cytochrome oxidase subunits 1–3, components of complex I (ND1 and ND2)) could have occurred with PPARδ gene silencing. Consequently, the latter could have reduced the mitochondrial pyruvate derived MAPR. Indeed, the expression of PGC-1*α* markedly decreased in PPARδ deficient heart, concomitant with a decreased mitochondrial DNA copy number [33]. Moreover, glucose oxidation rates were markedly depressed in the cardiomyocytes of PPARδ knockout hearts [33].

Interestingly, similarities in the pattern of PDC activity and its relative function after *PPARδ* and *FOXO1* silencing were observed. Therefore, we explored whether an interaction between PPARδ and FOXO1 may exist. The protein results showed that *PPARδ* silencing reduced total FOXO1 protein, but FOXO1 silencing did not change the expression levels of PPARδ protein. The latter finding along with the reported greater significance of *PPARδ* silencing on rescuing PDC activity and reducing media lactate accumulation, even in the presence of PAL compared with FOXO1 silencing, suggests that PPARδ may be an upstream modulator of FOXO1.

Study limitation. During the experiments undertaken in the present study, *PPARδ* siRNA gene silencing in the presence of PAL depressed pyruvate derived MAPR responses, contrasting with those from *PPARα* and *FOXO1*. This finding is not in accordance with a previous in vivo finding [29]. Although we provided a cautious interpretation earlier based on evidence from the literature of this in vitro finding [30,32,33], it should be acknowledged that this could have also been attributable to the in vitro results not being extrapolatable to an in vivo environment.

## 5. Conclusions

In summary, the findings of this molecular investigation suggest that transcription factors PPARδ, FOXO1 and PPARα play, in decreasing order, essential roles in cellular fuel selection in skeletal muscles by downregulating glucose-derived pyruvate oxidation during conditions of chronically elevated levels of FFAs. It appears that this occurs due to the ability of PPARδ and FOXO1 to increase muscle PDK4 protein levels and through PPARα allosteric inhibition of flux through PDC reaction. Furthermore, since *PPARδ* seems to control FOXO1 expression, this may reflect an important role for *PPARδ* in preventing glucose oxidation under conditions of increased lipid availability.

## Figures and Tables

**Figure 1 biology-10-01098-f001:**
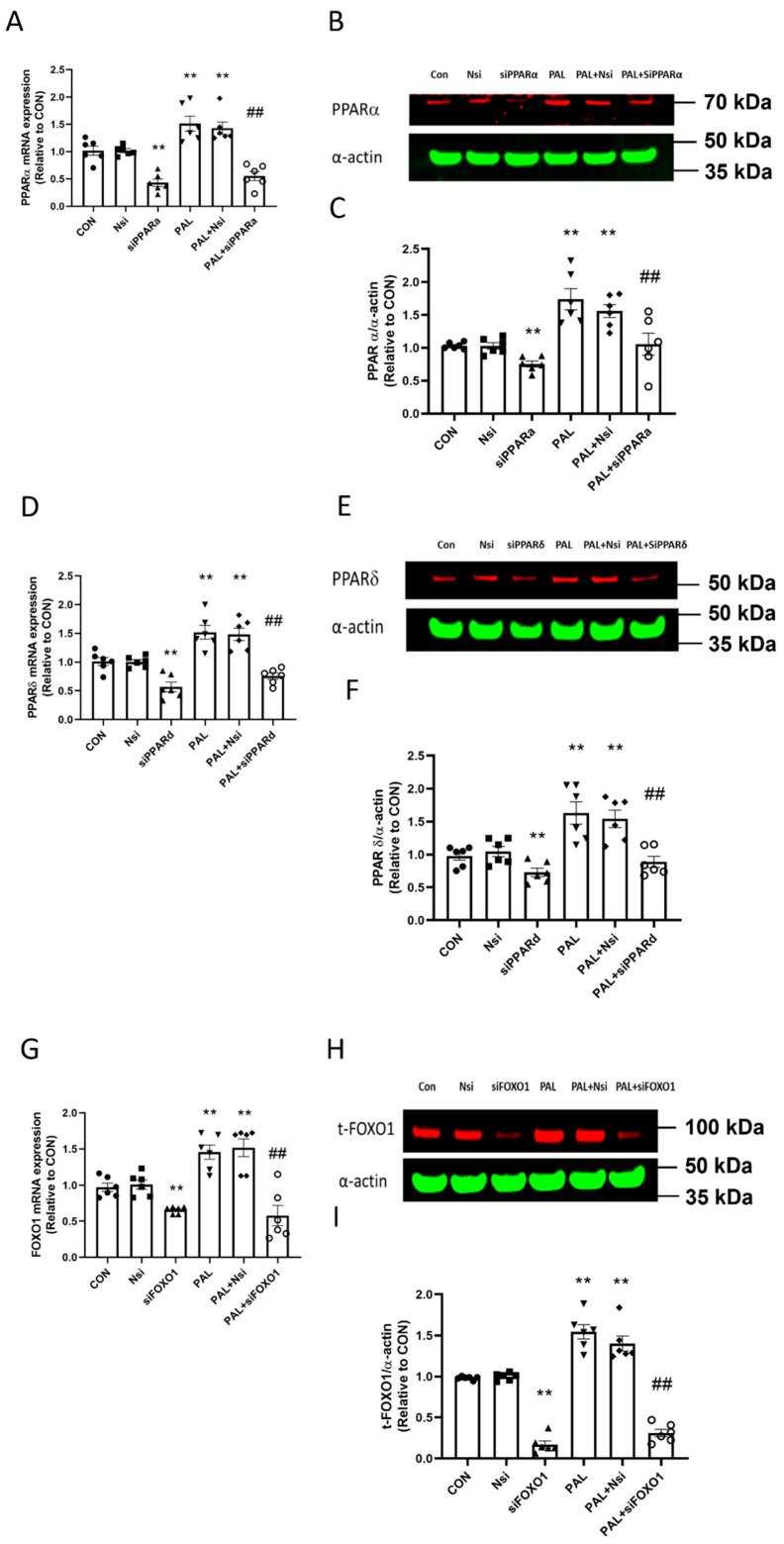
Validation of individual PPARα, PPARδ and FOXO1 siRNA gene silencing at transcriptional (**A**,**D**,**G**, respectively) and translational levels (**B**,**C**,**E**,**F**,**H**,**I**, respectively) in C2C12 myotubes in the absence (**left panels**) or the presence of palmitate (PAL; **right panels**). Significantly different from control without PAL (CON, ** *p* < 0.01). Significantly different from CON + PAL (^##^
*p* < 0.01). Data represent mean ± SEM of 6 individual experiments for each intervention group. Nsi—silencer negative control. Full WB can be found in Appendix A.

**Figure 2 biology-10-01098-f002:**
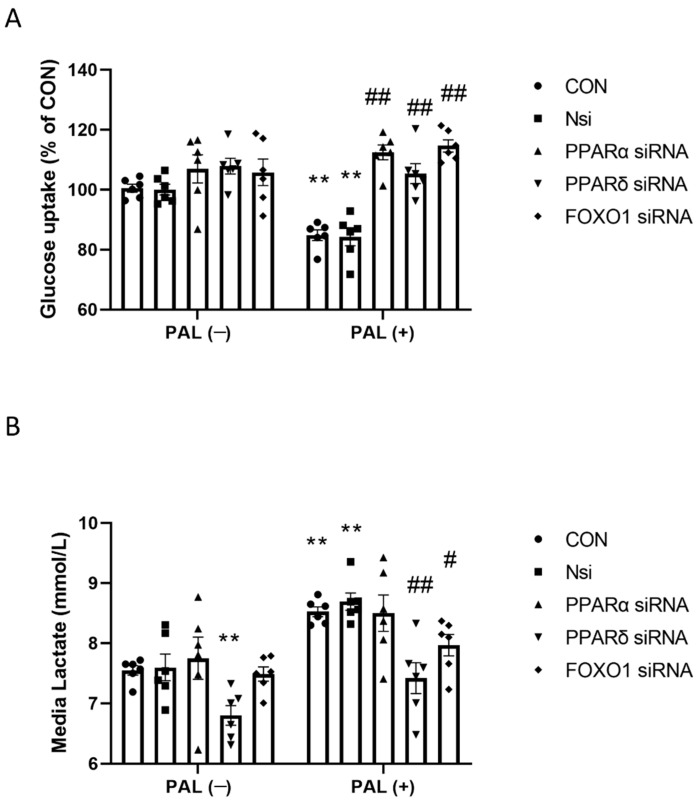
Glucose uptake (**A**) and cell media lactate concentration (**B**) in C2C12 myotubes after individual PPARα, δ and FOXO1 siRNA gene silencing in the absence (**left panels**) or the presence of palmitate (PAL; **right panels**). Significantly different from control without PAL (CON, ** *p* < 0.01). Significantly different from CON + PAL (^#^
*p* < 0.05; ^##^
*p* <0.01). Data represent mean ± SEM of 6 individual experiments for each intervention group. Nsi—silencer negative control.

**Figure 3 biology-10-01098-f003:**
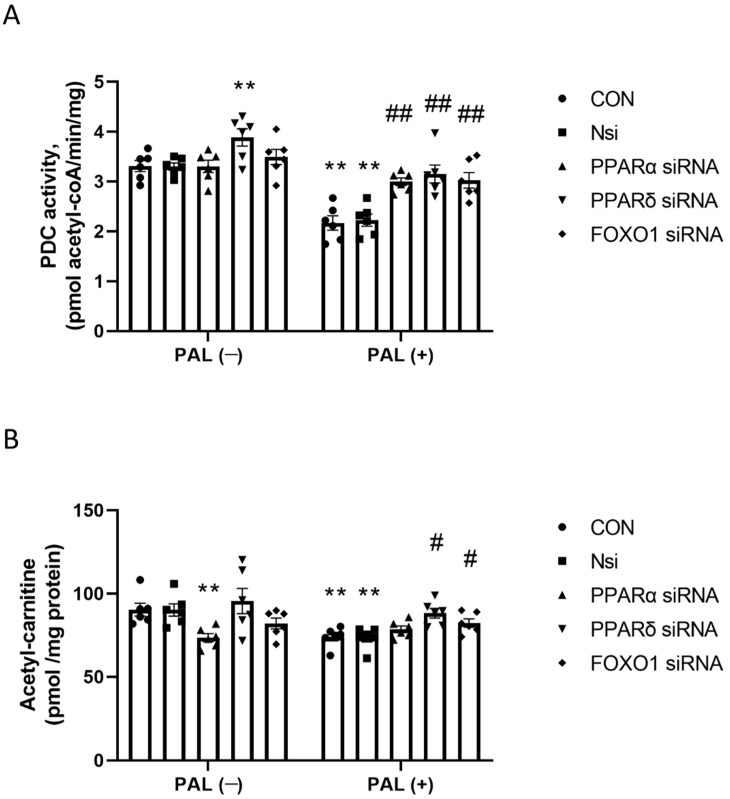
PDC activity (**A**) and acetylcarnitine accumulation (**B**) in C2C12 myotube after individual PPARα, PPARδ and FOXO1 siRNA gene silencing in the absence (**left panels**) or the presence of palmitate (PAL; **right panels**). Significantly different from control without PAL (CON, ** *p* < 0.01). Significantly different from CON + PAL (^#^ *p* < 0.05; ^##^ *p*< 0.01). Data represent mean ± SEM of 6 individual experiments for each intervention group. Nsi—silencer negative control.

**Figure 4 biology-10-01098-f004:**
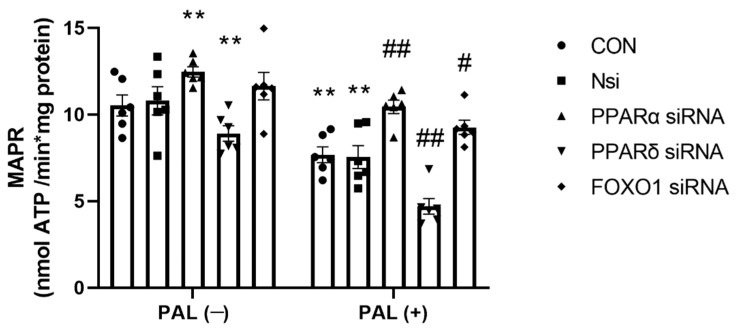
Maximal mitochondrial ATP production rates (MAPR) from pyruvate + malate in C2C12 myotube after individual PPARα, PPARδ and FOXO1 siRNA gene silencing in the absence (**left panels**) or the presence of palmitate (PAL; **right panels**). Significantly different from control without PAL (CON, ** *p* < 0.01). Significantly different from CON + PAL (^#^
*p* < 0.05; ^##^
*p* < 0.01). Data represent mean ± SEM of 6 individual experiments for each intervention group. Nsi—silencer negative control.

**Figure 5 biology-10-01098-f005:**
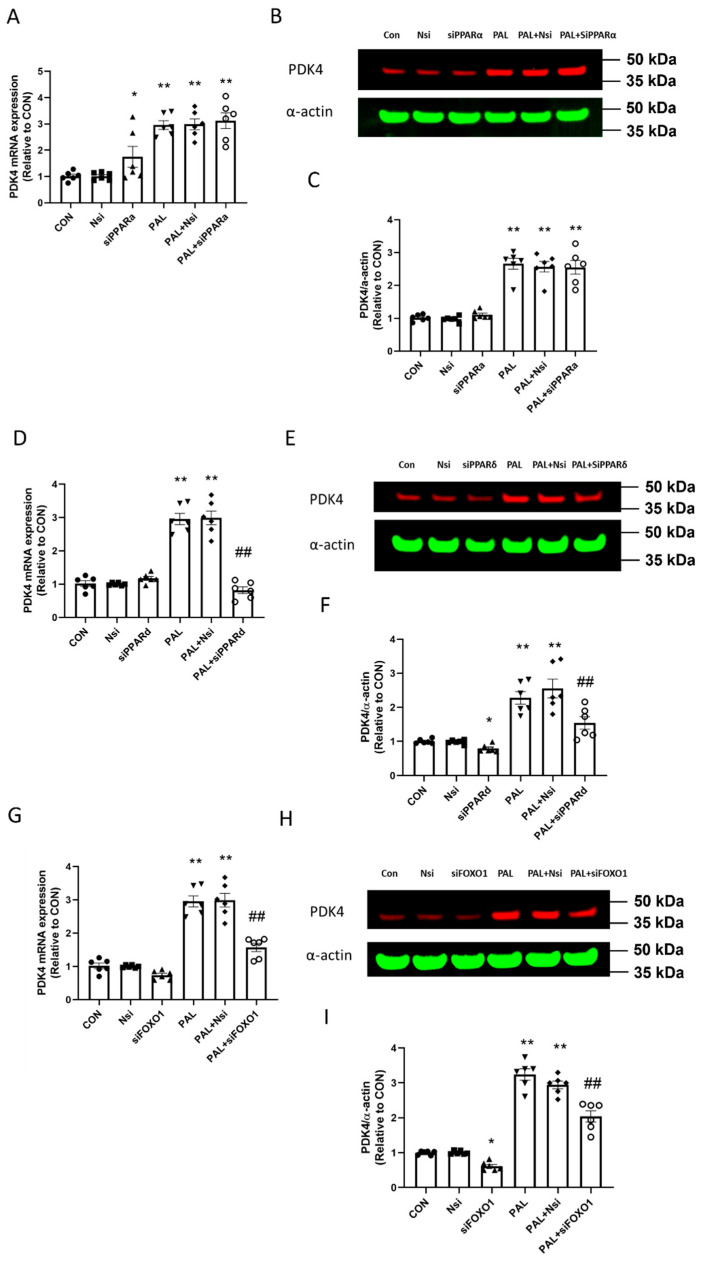
PDK4 gene (**A**,**D**,**G**) and protein expression (**B**,**C**,**E**,**F**,**H**,**I**) in C2C12 myotubes after individual PPARα, PPARδ and FOXO1 siRNA gene silencing in the absence (**left panels**) or the presence of palmitate (PAL; **right panels**). Significantly different from control without PAL (CON, * *p* < 0.05, ** *p* < 0.01). Significantly different from CON + PAL (^##^ *p* < 0.01). Data represent mean ± SEM of 6 individual experiments for each intervention group. Nsi—silencer negative control.

**Figure 6 biology-10-01098-f006:**
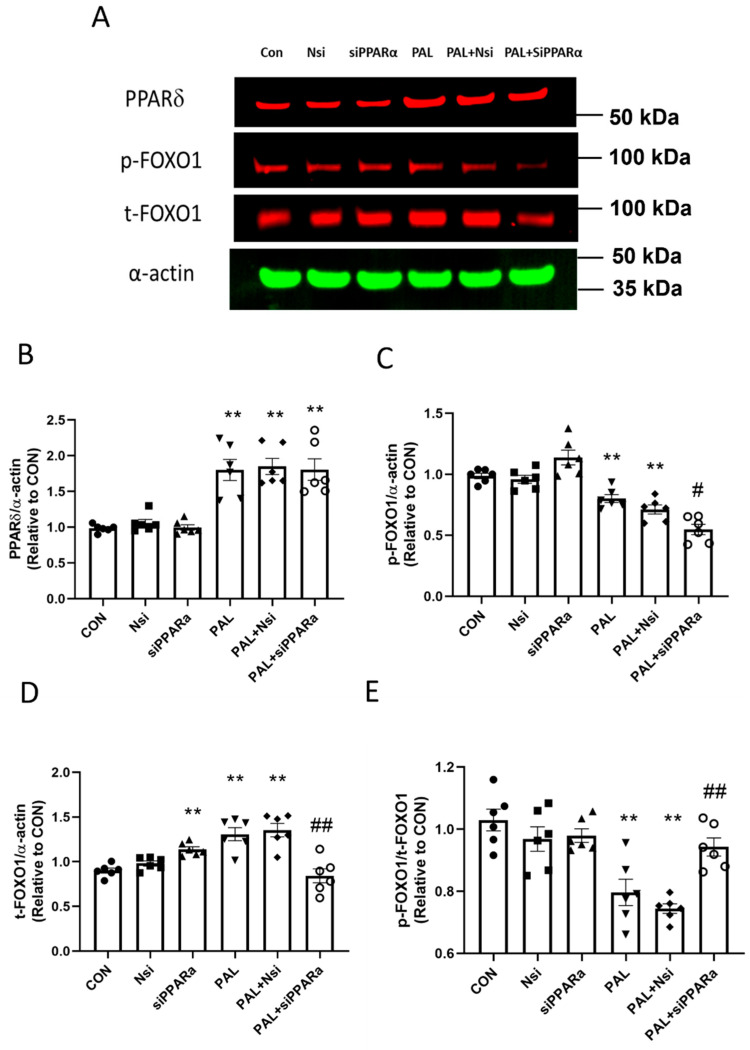
PPARδ (**A**,**B**) and FOXO1 (**A**,**C**,**D**,**E**) protein expression in C2C12 myotubes after PPARα siRNA gene silencing in the absence (**left panels**) or presence of palmitate (PAL; **right panels**). Significantly different from control without PAL (CON, ** *p* < 0.01). Significantly different from CON + PAL (^#^
*p* < 0.05; ^##^
*p* < 0.01). Data represent mean ± SEM of 6 individual experiments for each intervention group. Nsi—silencer negative control.

**Figure 7 biology-10-01098-f007:**
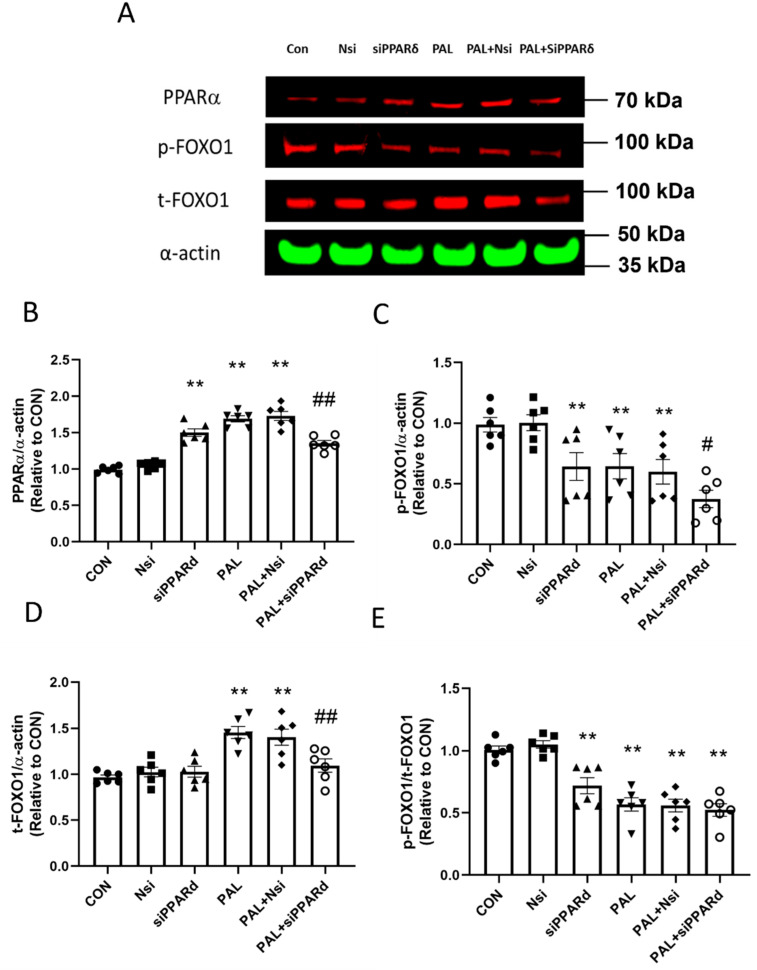
PPARα (**A**,**B**) and FOXO1 (**A**,**C**,**D**,**E**) protein expression in C2C12 myotube after PPARδ siRNA gene silencing in the absence (**left panels**) or the presence of palmitate (PAL; **right panels**). Significantly different from control without PAL (CON, ** *p* < 0.01). Significantly different from CON + PAL (^#^
*p* < 0.05; ^##^
*p* < 0.01). Data represent mean ± SEM of 6 individual experiments for each intervention group. Nsi—silencer negative control.

**Figure 8 biology-10-01098-f008:**
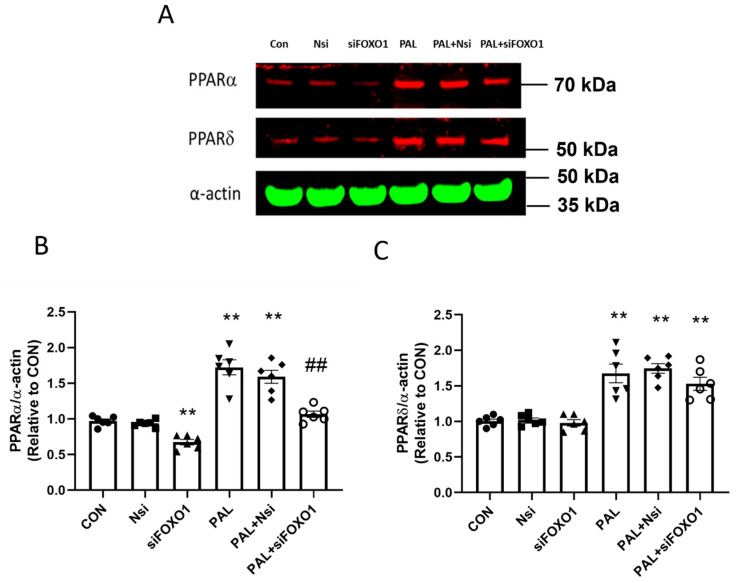
PPARα (**A**,**B**) and PPARδ (**A**,**C**) protein expression in C2C12 myotubes after FOXO1 siRNA gene silencing in the absence (**left panels**) or the presence of palmitate (PAL; **right panels**). Significantly different from control without PAL (CON, ** *p* < 0.01). Significantly different from CON + PAL (^##^
*p* < 0.01). Data represent mean ± SEM of 6 individual experiments for each intervention group. Nsi—silencer negative control.

## Data Availability

Not applicable.

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
