# Peer review of "PPARα, δ and FOXO1 Gene Silencing Overturns Palmitate-Induced Inhibition of Pyruvate Oxidation Differentially in C2C12 Myotubes"

_biology, 2021, doi:10.3390/biology10111098_

Round 1
Reviewer 1 Report
The manuscript (Biology) entitled “PPARα, δ and FOXO1 gene silencing overturns differentially 2 palmitate-induced inhibition of pyruvate oxidation in C2C12 3 myotubes” is by Hung-Che Chien et al. The authors claimed that the mechanism by which palmitate (PAL) downregulates pyruvate dehydrogenase complex (PDC)-mediated reduction in carbohydrate (CHO) oxidation in muscle occurs primarily through an increase in PPARδ and FOXO1 mediated raise in PDK4 protein,
Major Comments:
- Please note, carbohydrates include a variety of molecules. However, PAL inhibits the use of glucose in skeletal muscle cells. Therefore, please be precise.
- Extensive language editing is needed. For example, in line 47 on page 2/17, what did you mean “…carbohydrates (CHO) muscle use…”?
- The concentration pf PAL and length of treatments in the presence of PAL were not described. This is unacceptable.
- PAL is only one saturated fatty acid. Other mono- and poly-unsaturated fatty acids should be tested.
- In Figure 2B, in the presence of PAL, the lactate production was actually increased in the control group. On the other hand, the glucose uptake (Figure 1A) was reduced in the presence of PAL. If glucose uptake is inhibited in the presence of PAL, where did the lactate come from? Please explain.
- Silencing PPARα, δ or FOXO1 only affect the PDC activity slightly. Please measure the glucose usage rates, not the glucose uptakes.
- Since specific ligands for PPARα, and δ are available, additional experiments must be included to measure the PDC activity, lactate production, glucose usage in the presence or absence of these ligands. In so doing, you can decipher the specific roles of these transcription factors in the regulation of muscle cell glucose metabolism. The mRNA and protein levels of PDK4 and PDC complexes should be included as well.
Author Response
Reviewer 1.
The manuscript (Biology) entitled “PPARα, δ and FOXO1 gene silencing overturns differentially 2 palmitate-induced inhibition of pyruvate oxidation in C2C12 3 myotubes” is by Hung-Che Chien et al. The authors claimed that the mechanism by which palmitate (PAL) downregulates pyruvate dehydrogenase complex (PDC)-mediated reduction in carbohydrate (CHO) oxidation in muscle occurs primarily through an increase in PPARδ and FOXO1 mediated raise in PDK4 protein,
Major Comments:
Thank the reviewer for his/her helpful comments.
- Please note, carbohydrates include a variety of molecules. However, PAL inhibits the use of glucose in skeletal muscle cells. Therefore, please be precise.
We have replaced the term carbohydrates (CHO) with either glucose or glucose-derived pyruvate term.
2. Extensive language editing is needed. For example, in line 47 on page 2/17, what did you mean “…carbohydrates (CHO) muscle use…”?
For better clarity, we have replaced the “…carbohydrates (CHO) muscle use…” with “a blunting of the switch in muscle fuel use from fat to glucose upon feeding or insulin-stimulated conditions.”
Also, the manuscript underwent extensive language editing.
3. The concentration pf PAL and length of treatments in the presence of PAL were not described. This is unacceptable.
We apologise for the omission. The concentration of the working palmitate used in the experiments and a complete description of the sequence of the events are now presented in the Materials and Methods section text. It reads:
Once the myoblasts reached 80-100% confluency (after four days), they were differentiated into myotubes using a differentiating media containing high glucose DMEM, 2% Donor Horse Serum and 1% Pen/Strep. Two days into differentiation (day 6), the myotubes were transfected with 100 nM siRNA and 3 μL of Lipofectamine (ThermoFisher, Loughborough, UK) in an antibiotic-free medium in a 6-well according to the manufacturer’s instructions and allowed to grow for an additional two days. At the end of day 8, the media was removed and standard media (see above) with or without 500 μM palmitate was added to the cells, which were allowed to grow for a further 24 hr. At the end of day 9, the media was removed and the myotubes were harvested and used for different experimental measurements, as will be further described. The same concentrations of non-specific siRNA (scramble) with Lipofectamine were transfected in parallel plates as controls. The cells were throughout all phases and experiments constantly incubated at 37°C in 5% CO2.
4. PAL is only one saturated fatty acid. Other mono- and poly-unsaturated fatty acids should be tested.
We would agree with the reviewer, but the scope of the present manuscript was to challenge glucose-derived pyruvate oxidation in the presence of palmitic acid (PAL), which is the most prevalent saturated fatty acid (16:0) in human tissues. On average, a 70-kg man is made up of 3.5 kg of PAL. PAL is also a significant component of dietary meat and dairy products (50–60% of total fats). (https://www.ncbi.nlm.nih.gov/pmc/articles/PMC5682332/)
5. In Figure 2B, in the presence of PAL, the lactate production was actually increased in the control group. On the other hand, the glucose uptake (Figure 1A) was reduced in the presence of PAL. If glucose uptake is inhibited in the presence of PAL, where did the lactate come from? Please explain.
The reviewer is correct that glucose uptake with palmitate (Fig. 2B right panel) was 18% lower than without palmitate (Fig. 2A left panel), but the uptake was not entirely abolished. In the face of reduced glucose uptake, myotube metabolism will increase the reliance on glycogenolysis mediated pyruvate formation. Yet, since both PDC activity and pyruvate mediated mitochondrial ATP production was also reduced with palmitate (Fig. 3A, right panel and Fig. 4 right panel, respectively), the myotubes will also increase the reliance on anaerobic ATP production, which explains the greater rate of lactate accumulation (from pyruvate) in the control with palmitate.
6. Silencing PPARα, δ or FOXO1 only affect the PDC activity slightly. Please measure the glucose usage rates, not the glucose uptakes.
Mitochondria oxidise only glucose-derived pyruvate as the glucose molecules do not enter mitochondria; therefore, the glucose oxidation rate is twice that of pyruvate. The results reported in Fig. 4 as maximal mitochondrial pyruvate derived ATP production rates fully represent glucose oxidation rates and have the most significant meaning to the manuscript's hypothesis and narrative.
If somebody wishes to estimate glucose usage rate, this can be calculated by totalling the semi-sum of the mitochondrial ATP production and lactate accumulation rates. However, the reviewer will appreciate that these two processes are not linear throughout the experiments, as the aerobic glucose oxidation is waning over time. In contrast, that of glucose anaerobic use will increase.
7. Since specific ligands for PPARα, and δ are available, additional experiments must be included to measure the PDC activity, lactate production, glucose usage in the presence or absence of these ligands. In so doing, you can decipher the specific roles of these transcription factors in the regulation of muscle cell glucose metabolism. The mRNA and protein levels of PDK4 and PDC complexes should be included as well.
We have already published several works where we have extensively examined the reviewer’s suggestions, especially those related to PDC activity and lactate production. Please see the latest publication (https://www.mdpi.com/1422-0067/22/18/9775), which highlights these results in the form of a review.
Reviewer 2 Report
The authors have now point-by-point addressed all the major and minor points of this reviewer and provided the original western blots shown in the study. The manuscript is now substantially improved as compared to the initial one and the story is appropriately discussed and interpreted. Hence, no further revision is needed and the article can be published, if the editors of Biology agree, in its current form.
Author Response
We thank the reviewer for finding the revised manuscript deemed to be now acceptable for publication.
Reviewer 3 Report
In this manuscript, Chien et al. uncover mechanistic information helping determine how palmitate (as an FFA surrogate)leads to inhibition of carbohydrate oxidation in myoblasts.
The report is clear and the set of experiments is straightforward and uses well-established techniques and methods.
Some minor points of clarification to make the manuscript better
Authors should re-introduce abbreviations in the main body of the text, rather than introducing them in the abstract and then continuing to use them (for example, authors use MI in the introduction having only defined it in the abstract), and authors also need only introduce each abbreviation once. The authors define CHO, for example, in both of the first two paragraphs of the introduction.
Authors results section should stand on its own. For example the authors begin 2.1 with “The siRNA transfections were validated by real-tiem PCR and Western blotting techniques…”. Authors should instead describe that C2C12 myoblasts were grown in culture, subjected to differentiation conditions, and then transfected to knock down x, y, z… etc.
Line 89 “transcriptions” should be transfections.
Authors should use proper nomenclature (gene names and italicization) when talking about genes and transcripts specifically.
Authors should specify that their alpha-actin is specifically alpha-skeletal muscle actin.
Authors should subscript numbers appropriately for chemical formulas like CaCl2 or MgSO4
“Subsequently, the individual MAPRs were normalized for the protein content.” How was protein content determined?
Author Response
1. In this manuscript, Chien et al. uncover mechanistic information helping determine how palmitate (as an FFA surrogate)leads to inhibition of carbohydrate oxidation in myoblasts.
The report is clear and the set of experiments is straightforward and uses well-established techniques and methods.
Some minor points of clarification to make the manuscript better
We thank the reviewer for the highlighted issues meant to improve the clarity of the manuscript.
2. Authors should re-introduce abbreviations in the main body of the text, rather than introducing them in the abstract and then continuing to use them (for example, authors use MI in the introduction having only defined it in the abstract), and authors also need only introduce each abbreviation once. The authors define CHO, for example, in both of the first two paragraphs of the introduction.
We define now in the introduction all abbreviations, which were previously named in the abstract.
We have also omitted the CHO abbreviation throughout the text and replaced it with the term glucose.
3. Authors results section should stand on its own. For example the authors begin 2.1 with “The siRNA transfections were validated by real-tiem PCR and Western blotting techniques…”. Authors should instead describe that C2C12 myoblasts were grown in culture, subjected to differentiation conditions, and then transfected to knock down x, y, z… etc.
The current order of event description in the results section is the outcome that originated from a previous reviewer’s request.
Nevertheless, we felt that your suggestion could be addressed by introducing a description of the sequence of the events in the Material and Methods section as follows:
“Once the myoblasts reached 80-100% confluency (after four days), they were differentiated into myotubes using a differentiating media containing high glucose DMEM, 2% Donor Horse Serum and 1% Pen/Strep. Two days into differentiation (day 6), the myotubes were transfected with 100 nM siRNA and 3 μL of Lipofectamine (ThermoFisher, Loughborough, UK) in an antibiotic-free medium in a 6-well according to the manufacturer’s instructions and allowed to grow for a further two days. At the end of day 8, the media was removed and standard media (see above) with or without 500 mM palmitate was added to the cells, which were allowed to grow for a further 24 hr. At the end of day 9, the media was removed and the myotubes were harvested and used for different experimental measurements, as will be further described. The same concentrations of non-specific siRNA (scramble) with Lipofectamine were transfected in parallel plates as controls. The cells were throughout all phases and experiments constantly incubated at 37°C in 5% CO2. “
4. Line 89 “transcriptions” should be transfections.
We have replaced the term “transcriptions” with “transfections.”
5. Authors should use proper nomenclature (gene names and italicization) when talking about genes and transcripts specifically.
We have now italicized in the manuscript terms that refer to genes. In addition, a complete abbreviation list with the full name of all genes is reported at the end of the Discussion section.
6. Authors should specify that their alpha-actin is specifically alpha-skeletal muscle actin.
We have now named the antibody anti-skeletal muscle alpha-actin (ab28052).
Authors should subscript numbers appropriately for chemical formulas like CaCl2 or MgSO4
We have now appropriately under scripted the numbers in the chemical formulas.
7. “Subsequently, the individual MAPRs were normalized for the protein content.” How was protein content determined?
We have named the method used to determine protein concentration. The text now reads as follows: “Subsequently, the individual MAPRs were normalised for the protein content using the Bradford assay [18].”